# The Effect of Compressible Flow on Heat Transfer Performance of Heat Exchanger by Computational Fluid Dynamics (CFD) Simulation

**DOI:** 10.3390/e21090829

**Published:** 2019-08-25

**Authors:** Chao Yu, Sicheng Qin, Bosen Chai, Sen Huang, Yang Liu

**Affiliations:** 1School of Mechanical and Aerospace Engineering, Jilin University, Changchun 130022, China; 2Changchun Institute of Optics, Fine Mechanics and Physics, Chinese Academy of Sciences, Changchun 130033, China

**Keywords:** water cooled charge air cooler (WCAC), compressible flow, computational fluid dynamics (CFD), heat transfer

## Abstract

As a part of vehicle thermal management, water-cooled intercoolers play an important role in engine efficiency. The incompressible simulation model was usually applied to estimate the performance of water-cooled intercoolers. In this paper, the computational fluid dynamics (CFD) compressible model is taken to analyze more accurate prediction models. The rate of section change, heat exchange, and the surface friction coefficient are used as the comparison basis of the compressible flow model and incompressible model on the pressurized air side of the water-cooled intercooler. By comparing the simulation results of the air side, it was found that the compressible simulation is closer to the experimental value than the incompressible simulation. Compared with the experiment, the compressible model heat transfer maximum value of deviation is 6.5%, and the pressure loss maximum deviation is 7.5%. This provides guidance to optimize the design of heat exchangers, in order to save on costs and shorten development times.

## 1. Introduction

As a typical heat exchanger, intercoolers are used to cool high-temperature and high-pressure air from turbochargers. The use of an intercooler can greatly reduce pollutant emissions and improve the power performance of a diesel engine. [1]. 

With the improvement of computer performance, many scholars have carried out experimental and simulation research on intercoolers. Haitham Mezher et al. [2] experimentally analyzed the unsteady-state gas dynamics at the intake of a turbocharged engine, and proposed an optimal intake scheme for high-speed operation. This structure compensates for the lack of air pressure caused by the existing wave action at the intake. Jang-Won Seo et al. [3] studied the heat transfer and pressure drop characteristics of a surface heat exchanger when the Reynolds number changes. They found that the heat transfer growth as the flow rate increased. However, the pressure drop on the hot side decreased with the cold side Reynolds number. At the same time, they proposed the correlations of Nusselt number and friction factor. By fitting different kinds of fins in an intercooler, Chen et al. [4,5,6] obtained the correlations between heat transfer factors and friction factors of fins with different structures. The correlations can evaluate the heat transfer efficiency of fins better.

Zhang et al. [7] simulated the computational fluid dynamics (CFD) flow field of an intercooler and spotted reflux in the entrance area elbow, which was one of the main sources of pressure loss at the entrance. Momeni et al. [8] used the firefly algorithm for the multi-objective optimization of a marine diesel intercooler to find out the optimal working point. The results show that the overall cost and exergy destructions are reduced by 4.03% and 7.66%, respectively. The water pressure drop is decreased by 12.41% and the air pressure drop is reduced by 2.95%. Pierre Marty et al. [9] numerically simulated a large two-stroke diesel engine with its cooling and exhaust systems. Based on the simulation, they proposed a solution for energy and exergy saving which could be applied to marine engines.

The performance of the exhaust gas recirculation (EGR) intercooler system was also studied by researchers. Anuj Pal et al. [10] studied the emission characteristics of a single-cylinder four-stroke engine with an EGR intercooler. The results show that the use of an EGR intercooler can improve the emission performance. Billy G. Holland et al. [11] described a model for establishing a thermal fatigue life of EGR coolers by using conjugate transient CFD. The model can accurately simulate the metal temperature during heating and cooling cycles, and the accuracy of model verified by experiments. Jose J. Garcia [12] experimentally studied the low pressure cooling system of aluminum brazed EGR, air charge air cooler (ACAC), and water cooled charge air coolers (WCAC), and developed a test bed to evaluate the corrosion risk of acid solution in waste gas condensate. The experimental results were analyzed and the prospect of improving the corrosion resistance of heat exchanger components was introduced. S.S. Hoseini et al. [13] numerically investigated the heat transfer characteristics of shell and tube-type EGR intercoolers and stack type-EGR coolers. The results showed that the heat transfer efficiency of stack type-EGR coolers is higher than shell and tube-type EGR intercoolers. Riad Hadjb and Mahfoud Kadja [14] numerically simulated the heat transfer performance of EGR coolers applied to heavy-duty diesel engines. Compared the results, the EGR coolers composed of 19 tubes with helical baffles has the best performance in terms of cooling efficiency and pressure drop.

Researchers have also paid attention to the optimization of intercooler structures and thermal management. Mezher, Haitham and Miguaud, Jerome et al. [15] took the four-cylinder diesel engine as their research object. The results showed the reflection characteristics of the intake of water-cooled intercooler, and pointed out the optimal length of the pipeline between the intake manifold and intercooler. Pierre Salmon et al. [16] numerically studied the heat transfer, pressure drop and inhomogeneous flow. They provided numerical data to identify an optimum heat exchanger configuration for charge air coolers, which can be used in the design process of cooling systems. And a detailed approach for numerically determining the pressure drop and heat transfer characteristics of a heat exchanger operating in transitional flow regimes has also been provided. Xin Zhao et al. [17] proposed a conceptual design of a tubular two-pass crossflow intercooler architecture applied for a turbofan aeroengine application. They also simulated the internal flow by a porous media model for the intercooler tubes. Their work provided a case to evaluate the performance of intercooled aeroengines. M.T. Zegenhagen and F. Ziegler [18] experimentally analyzed an exhaust gas waste heat driven jet-ejector cooling system for charge air cooling of turbocharged gasoline engines. Based on the data, they determined the feasible cooling capacities and charge air temperatures. 

Most researchers attributed the deviation between simulated and experimental values to random errors or deviations due to excessive simplification of the model [19,20,21], but the method based on pressure solution neglects the change of density and the compressibility of fluid, which is also one of the important sources of deviation. 

In this paper, a compressible flow model of a heat exchanger based on fin wall friction, heat transfer and flow cross-section variation is established by studying the pressurized air side of a water-cooled intercooler of a wheel loader, and the prepared user defined functions (UDF) file is imported into Fluent to simulate the three-dimensional flow field. The reliability of this simulation is verified by comparing with simulation results based on pressure solution. Finally, the calculation results of the compressible model and the incompressible model are compared with the test data of the intercooler of a heat exchanger manufacturer to study the influence of compressible flow on the calculation of the heat exchanger. The results of this research will provide guidance for obtaining accurate prediction results, cost savings, and shortening development times.

## 2. Compressible and Incompressible Flow Model for Heat Exchanger

Many factors affect the gas flow in pipes, such as wall friction, mass force, mass conversion and gravity. Various factors have different influences on fluid flow. According to characteristics of the heat exchanger, friction resistance and heat transfer rate of air-side, the water-cooled intercooler will be studied to build a heat exchanger compressible and incompressible flow model.

The effect of compressible flow with heat exchange on heat exchanger calculations will be emphasized in this section. For building the heat transfer model, the compressible flow is simplified as a one-dimensional steady flow, friction-free heat exchange flow with no mechanical work shift, and (1)–(4) are introduced.
(1)$$\tau \left(\lambda \right)=1-\frac{k-1}{k+1}\lambda^{2}$$
(2)$$\pi \left(\lambda \right)={\left(1-\frac{k-1}{k+1}\lambda^{2}\right)}^{\frac{k}{k-1}}$$
(3)$$\pi \left(\lambda \right)={\left(1-\frac{k-1}{k+1}\lambda^{2}\right)}^{\frac{k}{k-1}}$$
(4)$$f\left(\lambda \right)={\left(\frac{2}{k+1}\right)}^{\frac{1}{k-1}}q\left(\lambda \right)z\left(\lambda \right).$$

In the formula, *z*(*λ*) is impulsive function, *k* is adiabatic exponent, *A_cr_* is critical section area, *λ* is dimensionless velocity coefficient.

Stagnation temperature rate can be made by the energy and momentum equation
(5)$$\frac{T_{b}^{*}}{T_{a}^{*}}=\left[\frac{z(\lambda_{a})}{z(\lambda_{b})}\right].$$

The temperature ratio is made by the substitution of density ratio, which is solved by the continuity equation
(6)$$\frac{T_{b}}{T_{a}}=\frac{T_{b}^{*}\tau \left(\lambda_{b}\right)}{T_{a}^{*}\tau \left(\lambda_{a}\right)}={\left[\frac{z(\lambda_{a})}{z(\lambda_{b})}\right]}^{2}\frac{\tau \left(\lambda_{b}\right)}{\tau \left(\lambda_{a}\right)}.$$

Stagnation pressure ratio and intensity of pressure of section a and b can be made from (4) and (7).
(7)$$\frac{p_{b}^{*}}{p_{a}^{*}}=\frac{f\left(\lambda_{a}\right)}{f\left(\lambda_{b}\right)}=\frac{p_{b}\pi \left(\lambda_{b}\right)}{p_{a}\pi \left(\lambda_{a}\right)}.$$

In the formula, *p** is stagnation pressure (Pa), *T** is stagnation temperature (K).

To make a comparison between the above models and solving value based on pressure (incompressible model), the pressure loss and heat transfer amount based on pressure solving are as follows:(8)$$\Delta P=\frac{1}{2}\rho fU_{min}^{2}$$
(9)$$Q=C_{p}\dot{m}\Delta t.$$

In the formula, f is the friction coefficient of heat exchanger, *U_min_* is the ratio of inflow velocity, *c_p_* is the specific heat at constant pressure J/(kg·K), $\dot{m}$ is the mass flow rate, Δ*t* is the temperature difference (K), and $\dot{m}$ is mass flow rate. The incompressible model is used the experimental correlation of Kays and London’s rectangular offset stripfin [22].

The two models are compared under the following conditions: The cold side is considered as the ideal one; the temperature of air side is 384.5 K and the pressure is 200 kPa; the fluid is dry air, so the adiabatic index is usually 1.4 in engineering; the circulation rate of heat exchanger core is 0.438; and the hot side heat transfer area is 2.06 m^2^.

The calculation results are shown in Figure 1. We can see that the pressure variation value is very small and in irregular fluctuations. It can be considered that the pressure variation of the intercooler caused by heat transfer in compressive flow calculations is very small. Therefore its pressure variation is not considered in the following thesis. The heat transfer amount in Figure 1 is the difference calculated by the compressible model and the incompressible model. It is rising with the flow rate increasing. In other words, the higher flow rate, the difference between compressible and incompressible models is the larger. It is clear that the calculation results in Figure 1 agree with the pressure, density and thermal variation rule in the subsonic cooling phase of the Rayleigh Curve.

## 3. Simulation Model and Boundary Conditions

### 3.1. Simulation Model 

In this section, compressible flow and incompressible flow are analyzed by Fluent for the hot side of the intercooler. The UDF, which is composed of pressure and temperature, is imported to do flow simulations in Fluent [23].

Considering the compressive flow model with many combined effects, and neglecting the gravity for complete air flow, the basic conservation laws can be written in the following forms: 

The energy equation can be written by this:(10)$$\delta Q-\delta W-dH_{a}=dH=C_{p}dT^{*}.$$

In this:$H=h+v^{2}/2+gz$, $dH_{a}=(H-H_{a})dm/\dot{m}$, *z* is the displacement of gravity. The formula is divided by *C_p_T*:(11)$$\frac{\delta Q-\delta W-dH_{a}}{C_{p}T}=\left(1+\frac{k-1}{2}Ma^{2}\right)\frac{dT^{*}}{T}.$$

The energy equation can be written:(12)$$\frac{dp}{p}+kMa^{2}\frac{dMa}{Ma}+\frac{kMa^{2}}{2}\left(\frac{dT}{T}+\frac{4fdx}{D}\right)+kMa^{2}\left(1-y\right)\frac{d\dot{m}}{\dot{m}}.$$

The state equation is:(13)$$\frac{dp}{p}=\frac{d\rho }{\rho }+\frac{dT}{T}.$$

The continuity equation is:(14)$$\frac{d\dot{m}}{\dot{m}}=\frac{d\rho }{\rho }+\frac{dV}{V}+\frac{dA}{A}.$$

The relationships between mach number and flow parameters, such as temperature *T*(K), section variation *dA*, hydraulic diameter *dA,* can be found by joining (15)–(18):(15)$$\frac{dMa}{Ma}=\frac{\left(1+\frac{k-1}{2}Ma^{2}\right)}{1-Ma^{2}}[\frac{kMa^{2}}{2}\left(\frac{4fdx}{D}\right)-\frac{dA}{A}+\frac{1+kMa^{2}}{2}\frac{dT^{*}}{T^{*}}]$$

In most projects, the geometric parameters of the heat exchanger, and entrance parameters are known conditions and the calculation step is set as Δx. In this article, mach numbers at arbitrary section in length are solved through the Runge-Kutta iterative method, then the numbers are brought into (6) and (7) to solve other flow parameters of sections with multiple combined effects. 

The standard *k-ε* model is used in this research. The model is based on the kinetic energy *k* equation, and then introduces an equation of turbulent dissipation rate *ε*. The *ε* is defined as:(16)$$\varepsilon =\frac{\mu }{\rho }\bar{\left(\frac{\partial u_{l}^{\prime }}{\partial x_{k}}\right)\left(\frac{\partial u_{l}^{\prime }}{\partial x_{k}}\right)}.$$

The turbulent viscosity can be expressed as a function of k and ε:(17)$$\mu_{t}=\rho C_{\mu }\frac{k^{2}}{\varepsilon }.$$

Therefore, the transport equation of the standard k-ε model is:(18)$$\left\{ \begin{array}{l} \frac{\partial \left(\rho \varepsilon \right)}{\partial t}+\frac{\partial \left(\rho ku_{i}\right)}{\partial x_{i}}=\frac{\partial }{\partial x_{j}}\left[\left(\mu +\frac{\mu_{t}}{\sigma_{\varepsilon }}\right)\frac{\partial \varepsilon }{\partial x_{j}}\right]+C_{1\varepsilon }\frac{\varepsilon }{k}\left(G_{k}+C_{3\varepsilon }G_{b}\right)-C_{2\varepsilon }\rho \frac{\varepsilon^{2}}{k}+S_{\varepsilon }\\ \frac{\partial \left(\rho k\right)}{\partial t}+\frac{\partial \left(\rho ku_{i}\right)}{\partial x_{i}}=\frac{\partial }{\partial x_{j}}\left[\left(\mu +\frac{\mu_{t}}{\sigma_{k}}\right)\frac{\partial k}{\partial x_{j}}\right]+G_{k}+G_{b}-\rho \varepsilon -Y_{M}+S_{k} \end{array}\right..$$

Where the kinetic energy caused by the average velocity gradient is:(19)$$G_{k}=\mu_{t}\left(\frac{\partial u_{i}}{\partial x_{j}}+\frac{\partial u_{i}}{\partial x_{i}}\right)\frac{\partial u_{i}}{\partial x_{j}}.$$

The turbulent energy generated by the buoyancy effect $G_{b}$ is:(20)$$G_{b}=\beta g_{i}\frac{u_{t}}{{\mathrm{Pr}}_{t}}\frac{\partial T}{\partial x_{i}}.$$
The following values are used for the standard k-ε model:C_1ε_ = 1.44, C_2ε_ = 1.92, C_3ε_ = 0.09, *σ_k_* = 1.0, *σ_ε_* = 1.3.

The hot side of the water-cooling intercooler of the loader is made up of 10 belts, the cold side is nine belts in this article. The solid model of the intercooler is shown in Figure 2. The core size is 120 mm × 300 mm × 110 mm. The type of flow of heat and cold fluid flow can be seen in Figure 2. Pressurized air flows in the horizontal direction, coolant flows in a U-shape, two times.

### 3.2. Boundary Conditions

The pitch of the fin in the hot side of the intercooler is 2.4 mm, and the height is 6.5 mm. The pitch of the fin in the cold side of the intercooler is 2.5 mm, and the height is 3 mm. The core used a porous medium model with the mesh divided by Gambit. The precision of model mesh is controlled to under 0.8, and mesh independence is checked in Figure 3. The final mesh amount of the model is 1.89 million. The CFD model is solved based on pressure and density respectively, and the physical property parameters of the pressurized air flow are shown in Table 1.

If the effect of the compressible flow is reflected better, the interior fins of the heat exchanger must be considered. With the precondition of grid quality guaranteed, the 10 hot-side heat radiations will have millions of grids. Obviously, the level of computer hardware must be high to realize the solving of this model. So this research is based on the 1/2 core modeling of the intercooler as shown in Figure 4. The grid quality is 0.35, and the grid number is 1.09 million. To prevent the disturbance caused by the sudden expansion of intercooler exits and entrances, the model neglects sudden contraction and expansion for intercooler exits and entrances. At the same time, we considered cold-side baffle as the ideal one. Without the condition of interfere among each radiating belt, the loss of pressure from each radiating belt will be in accordance with intercooler exits and entrances. The UDF is used in to make the simulation calculation of the compressible flow. The entrance parameter above is still used for the incompressible flow as the control group. 

## 4. Results and Discussion

### 4.1. Analysis of the Simulation Result

The cloud map of the scattered tropical center is shown in Figure 5, Figure 6 and Figure 7. Compared with the incompressible model, the velocity and temperature of the compressible model are higher, and more evenly distributed. When the gas flow rate in the intercooler exceeds 0.3 Ma, the compressible density of the gas increases and the pressure increases. At the same time, the temperature of the gas also increases.

Figure 8 and Figure 9 demonstrate the pressure and heat transfer curves respectively. Compared with the above calculation solutions, the difference value of the heat transfer amount of the two models in Fluent is much larger. From Figure 9, the compressible and incompressible difference value of the heat transfer amount is 0.6 kw when air inflow reaches 0.5 kg/s. At the same flow rate, the difference value of the heat transfer amount lowers to 2 kW, as shown in Figure 8. The reason is that the calculation solution aims to study the changes of flow parameters along the axis, while CFD simulation covers the changes of fluid along all aspects, which makes the accuracy and integrity of the three-dimensional simulation solution higher than that of the calculation solution. The pressure loss is shown in Figure 9, as well as the increasing speed of pressure loss when compressibility is considered. The main reason is that dynamic pressure goes down with the increase of density. Therefore, the frictional loss decreases. If the air is incompressible, the pressure loss and square of speed are in direct ratio to one another. 

### 4.2. Analysis on Experimental Comparison

In the experiment, the pressure and temperature sensor are installed on the heat side of intercooler, and a temperature sensor is installed on the entry and exit of the cold side. The data is collected by data acquisition instruments of DEWESoft (DEWE-43, Dewetron, Austria). Data post-processing is made using Matlab software (2017a, MathWorks, Natick, MA, USA). The condition of the hot side inflow is 2 bar, at 110 °C, the ethylene glycol coolant flow on the cold side is 100 L/min, at 40 °C.

The comparison of the compressible model, the incompressible model, and the experiment about heat transfer and pressure loss are shown in Figure 10 and Figure 11. The experimental deviation of pressure and heat transfer is shown in Table 2.

The deviation of the experimental value and simulation value from the incompressible model rises when the air flow exceeds 0.2 kg/s. The compressible curve still can be in high accordance with experimental values, which verifies the accuracy of the compressible model of the intercooler that was suggested above. All of these can be gained by conversion: The speed is 0.15 Ma when the flow is 0.2 kg/s; the speed is 0.1 Ma when the flow is 0.3 kg/s. The deviation of two simulating curves and experimental values can be neglected. When the speed is 0.15 Ma, the deviation of pre-experimental values of the incompressible model are at 5%. When the velocity reaches 0.2 Ma, the deviation of heat transfer and pressure loss of the incompressible model exceeds experimental values by 10%. The deviation of pressure loss even exceeds 30% when the velocity continues to increase and exceeds 0.3 Ma. The air flow of real projects is among 0.2~0.3 Ma, which means there will be some deviations when the incompressible model calculation is used. 

As shown in Figure 10, the deviation of compressible and incompressible simulation results and experimental values increases with the increase of speed. This is due to the fact that the simulation ignores the structural surface roughness and the idealized setting of the cold side of the intercooler. As shown in Figure 11, the compressible and incompressible simulation results have been higher than the experimental value and the deviation gradually increases with the speed. This is because the simulation ignores the influence of the inlet and outlet bulge expansion and the interference between the adjacent two layers of scatter tropics.

## 5. Conclusion

(1)Based on wall friction, heat exchange and variation of runner cross section, the compressible flow model of a heat exchanger is established. Compared with traditional solving methods based on incompressibility, the matching precision of the compressible flow model with the experimental value is higher. The largest deviation of heat exchange value and experimental value is 6.5%, the largest deviation of pressure loss is 7.5%, which guides for the optimization of the design of the heat exchanger.(2)When the flow velocity is low, both compressible and incompressible models can meet the requirements. But at high flow velocity, the accuracy of the compressible model is noticeably higher than that of the incompressible model.(3)In order to meet the demand of the engine for air intake, the compressible model can be used to predict the performance of the new structure of the intercooler more accurately. It also saves time and development costs. (4)In order to predict the performance of the intercooler more accurately, the simulation model should be further studied under multi-field coupling, in the future. A full size simulation of an intercooler will also need be carried out.

## Figures and Tables

**Figure 1 entropy-21-00829-f001:**
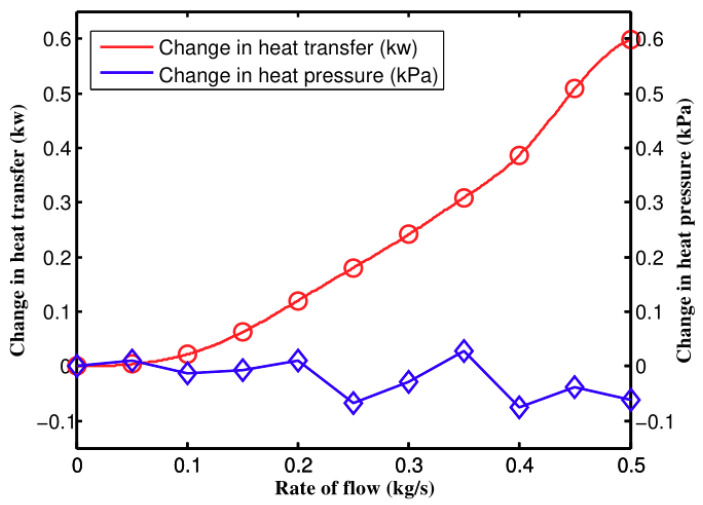
Pressure and heat transfer rate changes.

**Figure 2 entropy-21-00829-f002:**
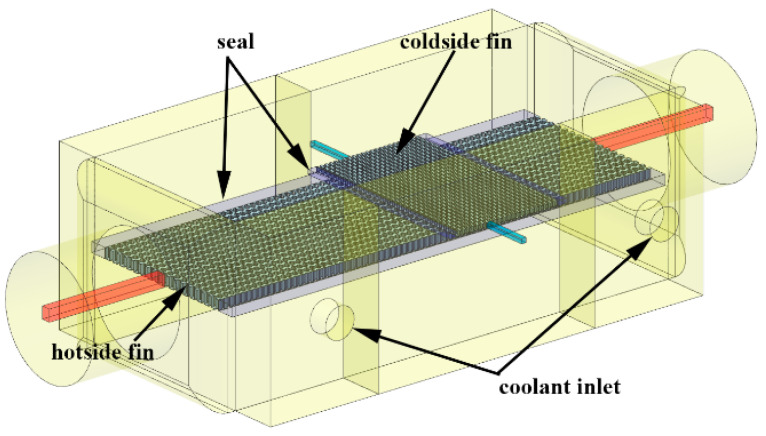
Model of a water-cooled charge air cooler.

**Figure 3 entropy-21-00829-f003:**
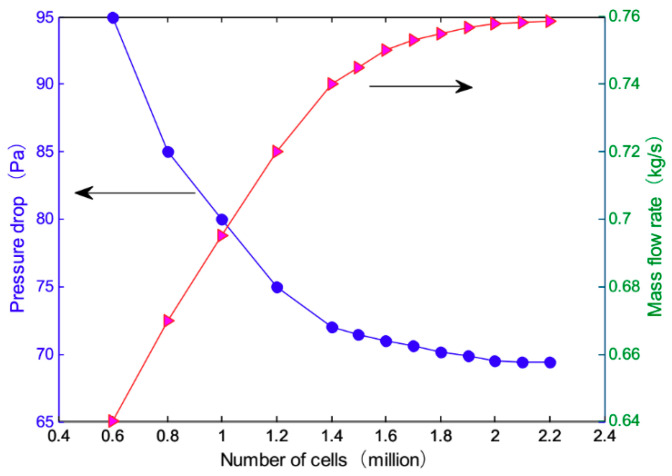
Mesh independence investigation.

**Figure 4 entropy-21-00829-f004:**
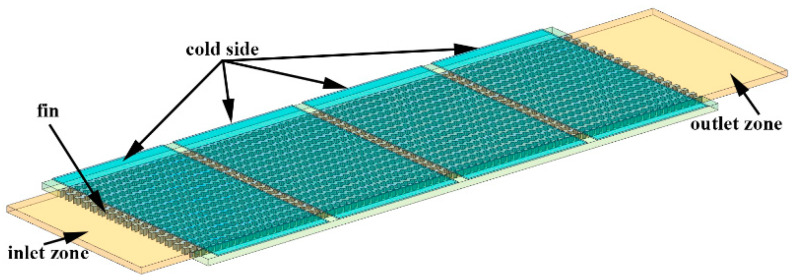
Hot side fin model.

**Figure 5 entropy-21-00829-f005:**
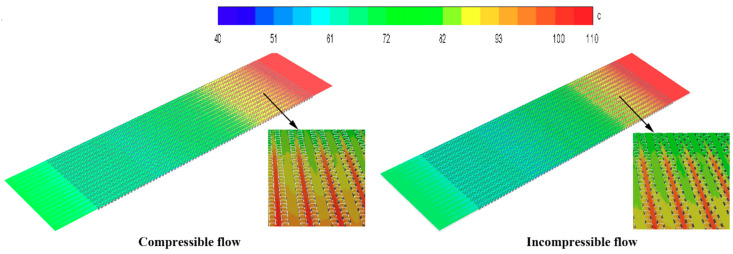
Temperature contours of the central section of the hot side.

**Figure 6 entropy-21-00829-f006:**
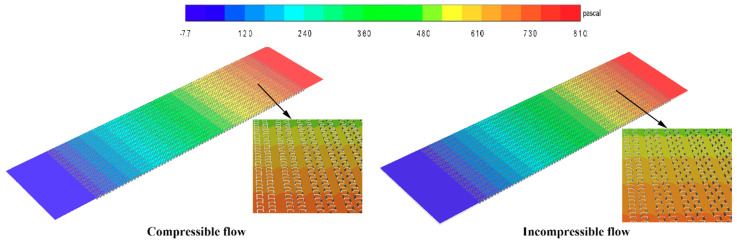
Pressure contours of the central section of the hot side.

**Figure 7 entropy-21-00829-f007:**
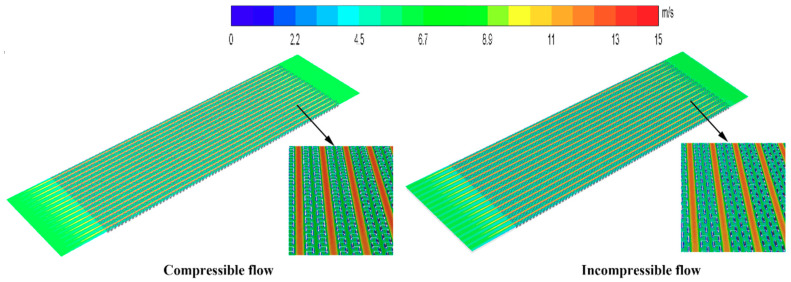
Velocity contours of the central section of the hot side.

**Figure 8 entropy-21-00829-f008:**
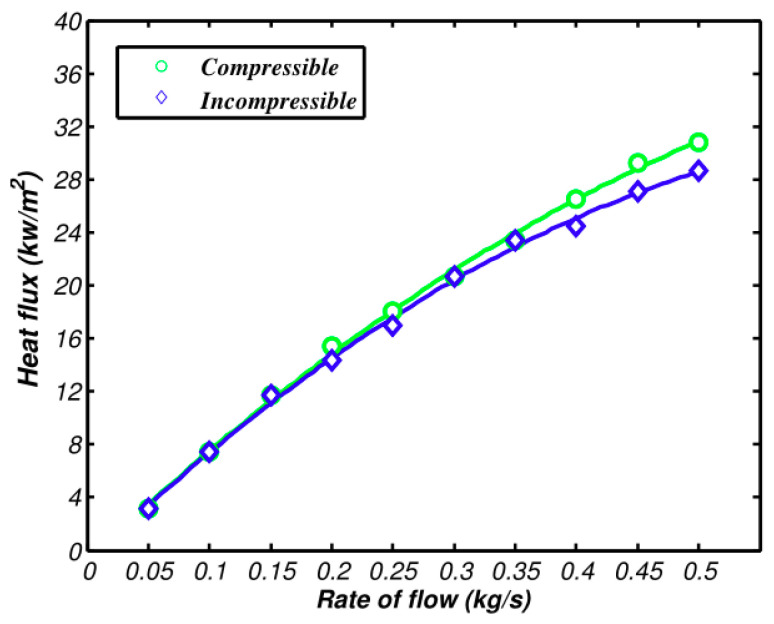
Comparison of heat transfer between compressible and incompressible flows.

**Figure 9 entropy-21-00829-f009:**
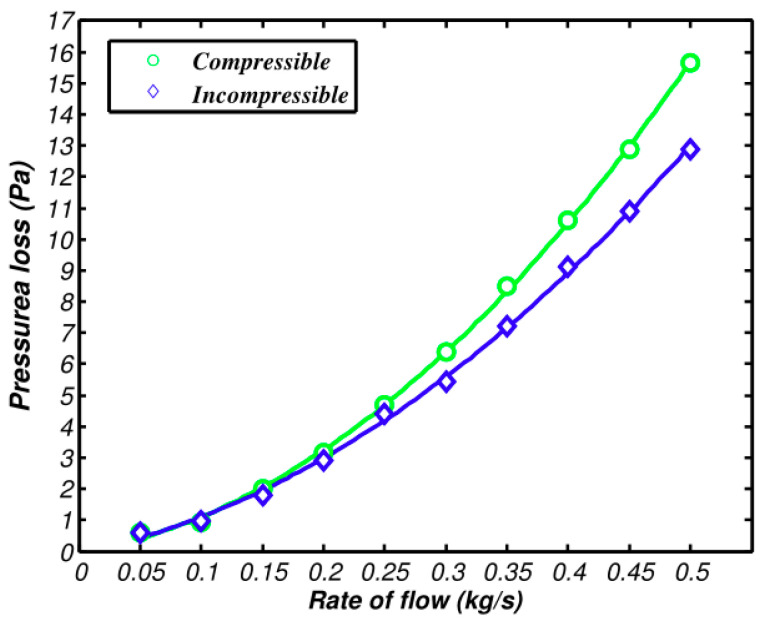
Comparison of pressure loss between compressible and incompressible flows.

**Figure 10 entropy-21-00829-f010:**
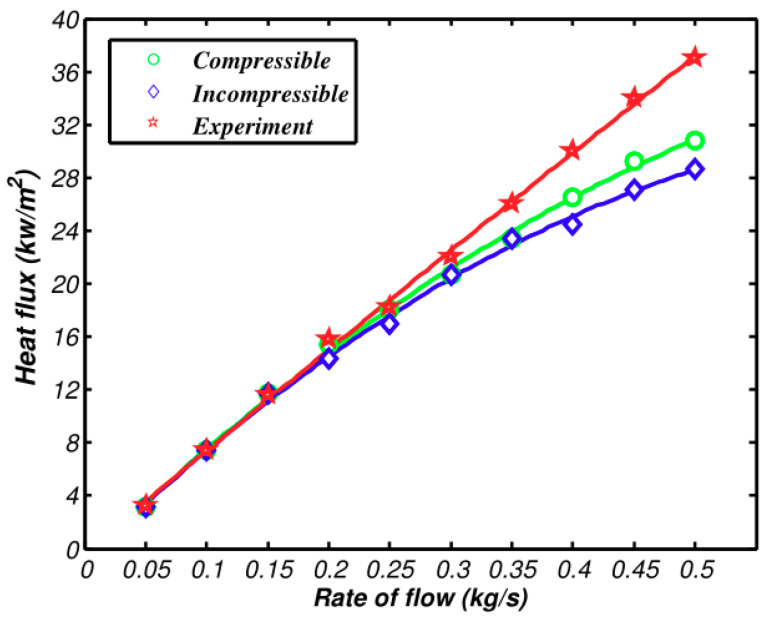
Comparison between simulation results and experimental results of heat transfer rates.

**Figure 11 entropy-21-00829-f011:**
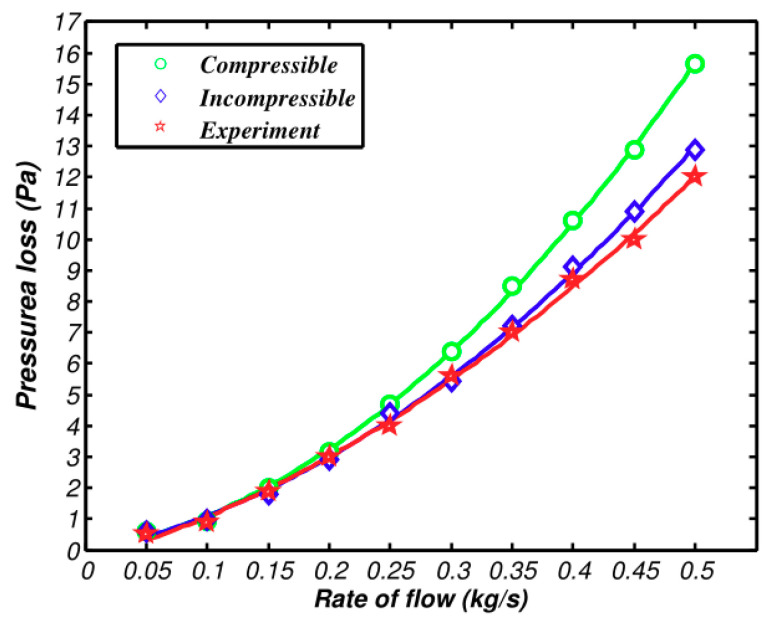
Comparison between simulation and experimental results of pressure loss.

**Table 1 entropy-21-00829-t001:** The physical properties of the pressurized air.

Density (kg/m^3^)	Specific Heat Capacity (kJ/kg·K)	Thermal Conductivity (W/m·k)	Dynamic Viscosity (Pa·s)
2.656	1.015	3.25 × 10^−2^	22.83× 10^−6^

**Table 2 entropy-21-00829-t002:** Deviation of heat transfer and pressure loss.

Flow Rate/(kg·s^−1^)	Heat Exchange Deviation/%		Pressure Loss Deviation/%	
	Compressible	Incompressible	Compressible	Incompressible
0.1	0.1	−0.1	3.4	5.2
0.2	0.7	−8.3	−3.2	5.0
0.3	−7.2	−14.3	−2.9	14.0
0.4	−4.2	−4.4	4.6	21.8
0.5	−6.5	−13	7.5	30.5

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
