# Peer review of "The Effect of Compressible Flow on Heat Transfer Performance of Heat Exchanger by Computational Fluid Dynamics (CFD) Simulation"

_entropy, 2019, doi:10.3390/e21090829_

Round 1

Reviewer 1 Report

General comments:

The topic of the paper is very interesting, within the scope of the journal, and worthy of investigation. The originality of the work is acceptable. The title is representative of the article contents.

However, I suggest that authors should take into consideration the following comments before the manuscript can be published in Entropy.

What is the authors’ contribution? Please add this part at the end of the manuscript.

Abstract:

Abstract introduces the readers with the subject of the investigation. However, the purpose of the study should be more highlighted and the broad context should be defined.

Please also check the sentence: “By comparing the simulation (…)” (lines 13-15). In my opinion it is not correct.

Introduction:

Although the introduction is very extensive, you do not define the broad context. Please add the paragraph at the beginning of this section to address this issue.

The importance and novelty of the study should be also more highlighted.

The names of the authors you refer to in the text are different than that in the references section (e.g. lines 31, 35, 41, 73). Please correct it. Please also do not use the names of all authors in the text (e.g. line 71 - change “Brian J. Luptowski, Oner Arici, John H et al.” on “Luptowski et al.”). Moreover, references should be indicated by numerals in square brackets ([1], [2]), not superscripts.

What’s more, almost half of the publications cited are more than ten years old. Please add new references (preferably from the last two years).

Compressible flow model for heat exchanger/ Numerical calculation of intercooler:

Line 100 – Please add numerals in the square bracket.

Please define ALL variables of the Equations and add units. It will be beneficial for the readers to add the nomenclature section.

Line 120 – Add this website to the references.

Section 2 and subsection 3.1 should be included in the Materials and Method section of the manuscript. In my opinion, you also should add the general subsection on cfd modeling.

Subsection 3.2 and 3.3 should be included in the Results and Discussion section. The discussion is insufficient. Please add deeper analysis.

Conclusion:

Please add the direction of future research.

References:

References relevant to Entropy should be added.

Author Response

Dear Editor:
  We would like to sincerely thank the reviewer for the very valuable and constructive suggestions. We have revised our manuscript from the suggestions. The following are our responses to the reviewers' comments:

To Reviewer #1:

Thank you for your question. We have modified or explained the following questions.

Abstract introduces the readers with the subject of the investigation. However, the purpose of the study should be more highlighted and the broad context should be defined.

The purpose of the study and broad context have been added as request. The corresponding content has been marked yellow.

Please also check the sentence: “By comparing the simulation (…)” (lines 13-15). In my opinion it is not correct.

The sentence is ambiguous and we have modified the sentence. The relevant content marked yellow.

Although the introduction is very extensive, you do not define the broad context. Please add the paragraph at the beginning of this section to address this issue.

The broad context has been defined as requested, and the relevant content marked yellow.

4、The importance and novelty of the study should be also more highlighted.

We have added more content to highlight the importance and novelty of the study in abstract and conclusion respectively.

5、The names of the authors you refer to in the text are different than that in the references section (e.g. lines 31, 35, 41, 73). Please correct it. Please also do not use the names of all authors in the text (e.g. line 71 - change “Brian J. Luptowski, Oner Arici, John H et al.” on “Luptowski et al.”). Moreover, references should be indicated by numerals in square brackets ([1], [2]), not superscripts.

The format has been modified as required. The names of the authors refer in text have checked and revised.

6、What’s more, almost half of the publications cited are more than ten years old. Please add new references (preferably from the last two years).

Ten references of the last five years were added.

7、Line 100 – Please add numerals in the square bracket..

There is no square bracket in line 100, but we check the format of this manuscript and revise some formatting errors.

8、Please define ALL variables of the Equations and add units. It will be beneficial for the readers to add the nomenclature section.

The variables of the equations and units have been added as request.

9、Line 120 – Add this website to the references.

The corresponding content has been add in reference.

10、Section 2 and subsection 3.1 should be included in the Materials and Method section of the manuscript. In my opinion, you also should add the general subsection on CFD modeling.

The Section 2 and subsection 3.1 have been a new section ‘Simulation model and boundary conditions’. The CFD modeling standard k-ε model has added as requested.

11、Subsection 3.2 and 3.3 should be included in the Results and Discussion section. The discussion is insufficient. Please add deeper analysis.

A new elaboration on the comparison between experiment and simulation has been added as the corresponding content marked yellow.

12、Please add the direction of future research.

The future research has been add in conclusion. In order to predict the performance of intercooler more accurately, the simulation model should be further studied under multi-field coupling, in the future. And the full size simulation of intercooler need be carried out. The corresponding content marked yellow.

13、References relevant to Entropy should be added.

   Ten references of the last five years were added, and some papers were published on Entropy.

Reviewer 2 Report

The title states that the manuscript contains CFD simulations, however the model presented in equations from 1 to 9 is a thermodynamic model, it is not clear what is the purpose of using the later model in the manuscript.

English of the manuscript should be thoroughly reviewed. There are many grammatical and spelling errors in all sections.

The abstract should follow the following structure: 1) State of the art, 2) Knowledge gap, 3) Objective, 4) Methodology, 5) Results, and 6) Conclusion

In the introductions, the authors are just mentioning that other researchers have analyzed. The authors did not write what other researchers have concluded. The Introduction should consist of a few paragraphs (perhaps no more than two) that define the context for the current work reported. How does this paper relate to what has been done previously? In the process it should point readers to publications to which they may need to refer in order to understand the motives for the current research.

In line 37, what does EGR mean?, the authors should define any acronym before use it.

In my opinion, the authors must present a validation/verification section so that you can verify that your cfd model is working correctly.

In line 92, replace "There are lots of factors that affect the gas flow in pipes" by "Many factors affect the gas flow in pipes"

In line 93, what do the authors mean with "heat exchanger caused by gravity"

What do the authors mean with "basic law of conversation"?

In line 102, replace the word "chapter" by “section”.

why is the adiabatic exponent equal to 1.4?, the authors must explain

In line 128, replace "heat exchange" by "heat transfer"

The methodology used by the authors is not clear, they must rewrite section 2. 

In the third column of table 1, the letter "w" must be uppercase

The conclusion are not meaningful,

The references must be updated, the authors are using only three references from the last three years, and the remaining references are somewhat old for a research paper.

Author Response

Dear Editor:
  We would like to sincerely thank the reviewer for the very valuable and  constructive suggestions. We have revised our manuscript from the suggestions. The following are our responses to the reviewers' comments:

To Reviewer #2:

Thank you for your question. We have modified or explained the following questions.

The title states that the manuscript contains CFD simulations, however the model presented in equations from 1 to 9 is a thermodynamic model, it is not clear what is the purpose of using the later model in the manuscript.

The paper use the equations 1~9 as the theoretical basis to prove the difference about the incompressible flow and compressible flow. Then we use the CFD simulation to compare the differences to Verify which model is more accurate and appropriate. This section may not be clear and has been modified.

English of the manuscript should be thoroughly reviewed. There are many grammatical and spelling errors in all sections.

It is our mistake about the grammatical and spelling errors in all sections. The English of the manuscript has been checked and revised by scholar whose first language is English.

The abstract should follow the following structure: 1) State of the art, 2) Knowledge gap, 3) Objective, 4) Methodology, 5) Results, and 6) Conclusion.

The abstract has been revised as the request.

4、In the introductions, the authors are just mentioning that other researchers have analyzed. The authors did not write what other researchers have concluded. The Introduction should consist of a few paragraphs (perhaps no more than two) that define the context for the current work reported. How does this paper relate to what has been done previously? In the process it should point readers to publications to which they may need to refer in order to understand the motives for the current research

The introductions has been revised, and ten new reference of the last five years were added.

5、In line 37, what does EGR mean?, the authors should define any acronym before use it.

The EGR means exhaust gas recirculation. And all the acronym has been defined.

6、In my opinion, the authors must present a validation/verification section so that you can verify that your CFD model is working correctly.

The CFD model verification section is included in 4.2 section. By the Fig.11 and Fig.12 showed the CFD model was available.

7、In line 92, replace "There are lots of factors that affect the gas flow in pipes" by "Many factors affect the gas flow in pipes".

The sentence has been revised as request. The corresponding content has been marked green.

8、In line 93, what do the authors mean with "heat exchanger caused by gravity"

The sentence has been revised as ‘Many factors affect the gas flow in pipes, such as wall friction, mass force, mass conversion and the gravity’.

9、What do the authors mean with "basic law of conversation"

The sentence has been revised as ‘The effect of compressible flow with heat exchange on heat exchanger calculation will be emphasized in this section’. The corresponding content has been marked green.

10、In line 102, replace the word "chapter" by “section”.

   The word "chapter" has been replaced by “section”.

11、Why is the adiabatic exponent equal to 1.4?, the authors must explain

The volume adiabatic index decreases with the increase of temperature and increases with the increase of pressure. When the pressure is less than 10MPa, the volume adiabatic index can be regarded as constant 1.4.When the pressure is greater than 10MPa, if the ratio of outlet pressure to inlet pressure is greater than 0.9, a fixed value of 1.4 is appropriate for the engineering volumetric adiabatic index.

12、In line 128, replace "heat exchange" by "heat transfer".

The word ‘heat exchange’ has been replaced by ‘heat transfer’ in the manuscript.

13、The methodology used by the authors is not clear, they must rewrite section 2.

 The section 2 has been rewritten as the corresponding content marked green.

14、In the third column of table 1, the letter "w" must be uppercase

   Thank you for your careful review. The letter "w" has been uppercase.

15、The conclusion are not meaningful

   The conclusion has been restated. The relevant content was marked by green.

16、The references must be updated, the authors are using only three references from the last three years, and the remaining references are somewhat old for a research paper.

Ten new reference of the last five years were added.

Round 2

Reviewer 1 Report

Thank you for considering my comments.

Reviewer 2 Report

The authors have attended all comments raised by the reviewer, I do recommend to accept the paper